# Profile of risk factors for Non-Communicable Diseases (NCDs) in a highly urbanized district of India: Findings from Puducherry district-wide STEPS Survey, 2019–20

Parthibane Sivanantham[1], Jayaprakash Sahoo[2], Subitha Lakshminarayanan[1], Zachariah Bobby[3], Sitanshu Sekhar Kar[1] *

1 Department of Preventive and Social Medicine, Jawaharlal Institute of Postgraduate Medical Education and Research (JIPMER), Puducherry, India, 2 Department of Endocrinology, Jawaharlal Institute of Postgraduate Medical Education and Research (JIPMER), Puducherry, India, 3 Department of Biochemistry, Jawaharlal Institute of Postgraduate Medical Education and Research (JIPMER), Puducherry, India

* drsitanshukar@gmail.com

## Abstract

### Introduction

Rapid urbanization and industrialization drives the rising burden of Non-Communicable Diseases (NCDs) worldwide that are characterized by uptake of unhealthy lifestyle such as tobacco and alcohol use, physical inactivity and unhealthy diet. In India, the prevalence of various NCDs and its risk factors shows wide variations across geographic regions necessitating region-specific evidence for population-based prevention and control of NCDs.

### Objective

To estimate the prevalence of behavioral and biological risk factors of NCDs among adult population (18–69 years) in the Puducherry district located in Southern part of India.

### Methodology

We surveyed adults using the World Health Organization (WHO) prescribed STEPwise approach to NCD surveillance (STEPS) during February 2019 to February 2020. A total of 2560 individuals were selected from urban and rural areas (50 clusters in each) through multi-stage cluster random sampling method. STEPS instrument was used to assess behavioral and physical measurements. Fasting blood sample was collected to estimate biochemical risk factors (Diabetes, Hypercholesterolemia) of NCDs.

### Results

Among men, alcohol use 40.4% (95% CI: 37.4–43.4) was almost twice higher compared to tobacco use 24.4% (95% CI: 21.7–26.9). Nearly half of the population was physically inactive 45.8% (95% CI: 43.8–47.8) and obese 46.1% (95% CI: 44–48.1). Hypertension and diabetes mellitus were present among one-third 33.6% (95% CI: 31.6–35.5) and one-fourth

**Data Availability Statement:** This study was conducted as part of a PhD project. As per the

institute's ethics guidelines, the ethics committee reserves the rights on the datasets which protects the privacy of research data for up to three years after completion of the studies. Therefore, the dataset cannot be shared at this point of time. In case of any further queries or clarifications regarding the institute's ethics guidelines on sharing of research data, the committee could be contacted through their official email: iechumanstudiesjipmer@gmail.com The data could be accessible from the JIPMER Ethics Committee (contact: iechumanstudiesjipmer@gmail.com) after three years of completion of the PhD project, for those who meet institute's criteria for access to confidential data.

**Funding:** Dr Sitanshu Sekhar Kar received the grant for the study. This work was funded by the Jawaharlal Institute of Postgraduate Medical Education and Research (JIPMER) Grant No. JIP/ Dean (R)/Intramural/Phs 1/2019-20 The url of Funder website: jipmer.edu.in/ The funder did not have any role in study design, data collection and analysis, decision to publish or preparation of manuscript.

**Competing interests:** The authors have declared that no competing interests exist.

26.7% (95% CI: 24.1–29.1) of the population which were significantly higher among men (37.1% vs 30.8% and 31.6% vs 23.2% respectively). Physical inactivity and overweight/obesity increased with increasing education levels. Tobacco and alcohol use was more common among men, whereas physical inactivity with obesity and hypercholesterolemia was higher among women.

## Conclusion

We found high prevalence of various NCDs and its risk factors among the adult population of Puducherry district.

## Introduction

Non-Communicable Diseases (NCDs) are the leading cause for morbidity and mortality worldwide, with three-fourth of deaths occurring in the low and middle-income countries like India [1]. Between the years 1990 and 2016, disease burden in India due to NCDs increased from 48% to 75%. Currently, three out of the top five causes for morbidity and mortality in the country are NCDs [2].

Rising burden of NCDs is due to rapid urbanization and social development occurring in the country over the past two decades. The change occurring in social structures transforms lifestyles of populations that are mainly characterized by increased adaptation of unhealthy diet, physical inactivity and tobacco use [1]. These modifiable risk factors precede the development of metabolic risk factors and then progress to NCDs in populations [3]. This progression when not intervened with prompt population-based health promotion interventions, causes a sizeable transition in the disease epidemiology of the population. A recent study estimated epidemiological transition levels (ETL) across states, which is a ratio of all-age disability adjusted life years (DALYs) due to communicable, maternal, neonatal, and nutritional diseases versus those due to NCDs and injuries combined. The study revealed wide variations in ETL across states as well as higher prevalence of NCD burden among the affluent states compared to less developed ones [2].

The geographical variation in NCD prevalence also informs the prevailing varying levels of NCD risk factors across the states. This warrants state-specific estimates for NCDs and its determinants, to deal effectively with the rising burden. Since 2010, with the launch of National Program for Prevention and Control of Cancer, Diabetes, CVD and Stroke (NPCDCS) and adoption of National NCD Monitoring framework [4, 5], several national and sub-national level surveys [6–9] have been conducted to estimate the prevalence of NCD risk factors across states in the country. However, except for tobacco, alcohol, overweight/obesity, and hypertension, state-specific estimates are currently unavailable for other NCD risk factors [10] which are crucial for devising population-specific interventions.

Puducherry is one of the highly urbanized union territories (UT) in the country [11]. It presents higher-middle ETL indicating higher NCD risk factor's burden [2]. However, no systematic NCD risk factors survey has been conducted in the area. Therefore, this study was undertaken in the district of Puducherry to estimate the prevalence of behavioral and biological risk factors using the World Health Organization (WHO) prescribed STEPwise Surveillance (STEPS) methodology.

## Methodology

### Study setting

Puducherry is one of the four districts of the union territory (UT) of Puducherry located along the eastern coast, of southern India. The population of Puducherry district is 0.98 million accounts for 76% of the total population of UT with majority (69.16%) residing in urban areas. The district has sex ratio of 1029, life expectancy (68.35 years), and literacy rate of 85.44% [12, 13]. Puducherry stands seventh in Human Development Index (HDI) compared to other Indian states [14]. This survey was conducted in both urban and rural areas of Puducherry district. Prevention and control activities such as population-based screening for NCDs and its risk factors, diagnostics, treatment initiation and follow-up are rendered in adherence to the NPCDCS guidelines through a network of public healthcare facilities, and nine medical colleges, and private hospitals.

### Study design and sample size

We conducted this NCD risk factors survey by following the STEPS guidelines during February 2019 to February 2020 [3, 9, 15]. We included 2560 adults (18–69 years) considering 50% prevalence of risk factors, 95% confidence interval and design effect of 1.5 using OpenEpi (version 3.01) to provide district wide gender and residence specific estimates with 90% response rate. Considering the population distribution of urban and rural to be 70:30, the sample was proportionately divided between urban (n = 1770) and rural (n = 790) areas.

### Sampling method

Multistage, stratified, geographically clustered sampling method was employed using Census 2011 data as the baseline. Three and two-staged sampling was done in urban and rural areas respectively. We included 50 clusters each from urban (out of 90 wards) and rural (62 revenue villages) in the first sampling stage. In urban areas, one census enumeration block (CEB) was chosen randomly from each selected ward in the second stage. From each selected CEB and village, 36 and 16 households were recruited respectively using systematic random sampling in the final sampling stage. One adult (18–69 years) member residing in the household for at least past six months was interviewed using KISH method [16]. Members of the household who had cognitive and physical impairment to a level that hindered them from understanding the questions and responding back were excluded from the sampling frame for selection of an adult.

### Data collection instrument

WHO STEPS (version 3.2) questionnaire was culturally adapted, translated to Tamil, and pilot tested before data collection. Risk factors were assessed in three steps. Behavioral risk factors such as tobacco use, alcohol consumption, physical inactivity, inadequate fruits and vegetables intake, were assessed in step one. Physical measurements for body mass index (BMI) and abdominal obesity were done in step two. Biochemical risk factors (fasting blood glucose and total cholesterol) were assessed in third step by collecting blood samples from alternate participants.

### Data collection procedure

In each CEB or village, the first household was chosen randomly and subsequently, the required number of households in the cluster was selected using systematic random sampling method. Data were collected from an eligible household member by a trained investigator

after obtaining informed written consent. All three survey steps were conducted at households. Non-response was considered when study participants were unavailable after two more visits were made at times convenient to them. The study was permitted by the Institute's (Jawaharlal Institute of Postgraduate Medical Education and Research) Ethics Committee *(JIP/IEC/2018/ 0246)*.

**Behavioral risk factors (Step 1).** All interviews were conducted by the first author after receiving adequate training in survey methodology. Demographic characteristics such as age, gender, residence, education, occupation including contact details were collected to begin with. Current use of tobacco (smoke and smokeless forms), alcohol (quantity), and intake of fruits and vegetables (quantity and frequency) were collected subsequently. Physical activity at work, travel and leisure activities were assessed using Global Physical Activity Questionnaire (GPAQ). Showcards were used to describe physical activity (type and intensity), standard drink (alcohol) and a serving of fruits and vegetables.

**Physical measurements (Step 2).** Height, weight, waist circumference (WC) and blood pressure (BP) were measured by following standard guidelines [3]. Participant's height was measured in standing posture with barefoot and light clothing. Measurements were done using portable stadiometer and electronic weighing scale at nearest 0.1cm and 100gm respectively. SECA constant tape was used for waist measurement. Three BP readings, each at three minutes apart were captured using electronic device *(OMRON, HEM 7120, Omron Corporation, Kyoto, JAPAN)*, and the last two readings were averaged to determine the BP status. Pregnant women were not subjected to physical measurements. All equipment was calibrated regularly before and during data collection.

**Biochemical measurements (Step 3).** Two trained field investigators having previous experience in collecting blood samples were recruited for carrying out Step 3 of the survey. They underwent training for three days that included an overview of all three steps, and procedure to be followed for collection and transportation of samples.

Fasting blood samples were collected from 50% of study participants from cubital region of arm in sitting position after explaining the procedure. Each participant was given a handout describing the need and purpose for overnight fasting (10–12 hours). Blood sample was collected in seated position in the living room of participant's household on the next day morning and was transported to the institute's biochemistry laboratory for estimation by maintaining appropriate cold chain.

Fasting blood glucose and total cholesterol were estimated using commercially available kits adapted to clinical chemistry autoanalyzer based on spectrophotometry *(Beckman Coulter Inc, Brea, California, USA)*. Glucose oxidase peroxide and cholesterol oxidase peroxide were used for the estimation at 520nm.

All information was collected in electronic version of the questionnaire loaded in the field data collection software ODK Collect (version 1.25.1).

## Operational definitions

Behavioral risk factors were determined based on the cut-offs recommended by STEPS guidelines [3]. Tobacco (smoke or smokeless) and alcohol use in last thirty days and one year respectively, was considered as current use. Alcohol use was assessed as Standard drinks (one standard drink = 100ml wine or 285ml bear or 30ml Spirit / Toddy / Arrack). Men and women consumed at least six and four standard drinks respectively, in at least one occasion during last 30 days, were considered to have harmful use of alcohol. Along with type of physical activity (work, travel and leisure), intensity levels (low, moderate and severe) were also captured. This was factored with the duration of such physical activities to derive metabolic

equivalent time (MET-minutes/week). Those had less than 600 MET, 600–1500 MET, and more than 1500 were classified to have low, moderate and high levels of physical activity, respectively. Behavioural assessment was based on self-report.

Overweight (23 to 24.99kg/m$^2$) and obesity ($\geq$25kg/ kg/m$^2$) were determined by following Asian cut-offs of BMI classification. WC $\geq$90cm for men and $\geq$80cm for women were regarded as abdominal obesity [17]. Systolic BP (SBP) of $\geq$140mm Hg or Diastolic BP (DBP) of $\geq$90mm Hg or currently on hypertension lowering drugs was considered as raised BP (hypertension) [18]. Diabetes mellitus (DM) was determined at Fasting plasma glucose (FPG) of $\geq$126 mg/dl or currently on anti-diabetic medications [19]. Hypercholesterolemia was defined as those having total cholesterol level of $\geq$200mg/dl [18] or currently on lipid-lowering drugs.

## Statistical analysis

Data cleaning and analysis were conducted using STATA (StataCorp LP, College Station, TX, USA). Continuous variables such as age, quantity of tobacco sticks and alcohol use, and categorical variables such as prevalence of various risk factors (tobacco and alcohol use, physical inactivity, raised BP, diabetes etc.) were summarized using mean (SD) and proportions respectively. Significant differences in risk factors prevalence between sub-groups of independent variables were determined by comparing 95% confidence interval.

The effect of clusters (villages/wards) on dependent variables was assessed using Intraclass Correlation Coefficient (ICC) and presence of statistically significant difference in the intercept only mixed-effects model compared to standard binary logistic regression. Dependent variables showing statistically significant improvement in the model fit, forward weighted adjusted prevalence ratio were determined using 'multilevel mixed-effects generalized linear model' under 'Poisson's regression'. Those variables showed non-significant improvement in model fit, weighted forward stepwise generalized linear modelling using 'Poisson's regression' was done to determine the adjusted prevalence ratio. The models were developed for each risk factor by keeping the risk factor as dependent variable and socio-demographic characteristics (gender, age group, education, marital status, occupation) as independent variables. In regression models, p$\leq$0.05 were considered statistically significant.

## Results

The response rate for the steps 1&2 and 3 were 2415/2560 (94.3%) and 1117/1208 (92%) respectively. Among the participants, mean (SD) age was 44.3 (14) years, women 1329/2560 (55%), and nearly half received at least secondary education 1173/2415 (48.6%) and also employed 1237/2415 (51.2%). Socio-demographic characteristics are provided in the Table 1.

The mean (CI) of various behavioral risk factors and physical measurements are given in Table 2. Fruits and vegetables intake and physical activity were significantly higher among men. The intake levels were significantly higher among urban population, whereas physical activity was significantly higher among rural population. BMI was significantly higher among women 25.8% (95% CI: 25.5–26).

The prevalence of current tobacco and alcohol use were 12.3% (95% CI: 11–13.7) and 18.5% (95% CI: 16.9–20) respectively, which were significantly higher among men. Alcohol use 40.6% (95% CI: 37.5–43.9) was almost twice higher compared to tobacco use 24.4% (95% CI: 21.7–26.9) among men. About half 45.8% (95% CI: 43.8–47.8) of the population was physically inactive and it was nearly twice higher among urban residents. Nearly nine out of ten people 86.8% (95% CI: 85.4–88.1) in Puducherry consumed fruits and vegetables in inadequate amounts (Table 3).

**Table 1. Socio-demographic profile of study participants (N = 2415).**

| Variables | Male (n = 1086) n (%) | Female (n = 1329) n (%) | Both Gender (N = 2415) n (%) |
|---|---|---|---|
| **Age categories (in years)** | | | |
| 18–44 | 538 (49.5) | 667 (50.2) | 1205 (49.9) |
| 45–69 | 548 (50.5) | 662 (49.8) | 1210 (50.1) |
| **Residence** | | | |
| Urban | 749 (69) | 932 (70.1) | 1681 (69.6) |
| Rural | 337 (31) | 397 (29.9) | 734 (30.4) |
| **Educational status** | | | |
| No formal education | 67 (6.2) | 223 (16.8) | 290 (12) |
| Less than primary | 121 (11.1) | 193 (14.5) | 314 (13) |
| Primary education | 287 (26.4) | 348 (26.2) | 635 (26.3) |
| Secondary/ Higher Secondary | 374 (34.4) | 360 (27.1) | 734 (30.4) |
| Graduation and above | 235 (21.6) | 204 (15.3) | 439 (18.2) |
| Refused to answer | 2 (0.2) | 1 (0.1) | 3 (0.1) |
| **Religion** | | | |
| Hindu | 1027 (94.6) | 1243 (93.5) | 2270 (94) |
| Christian | 38 (3.5) | 56 (4.2) | 94 (3.9) |
| Muslim | 21 (1.9) | 28 (2.1) | 49 (2) |
| Others | 0 | 2 (0.2) | 2 (0.1) |
| **Marital status** | | | |
| Never married | 228 (21) | 90 (6.8) | 318 (13.2) |
| Currently married | 831 (76.5) | 988 (74.3) | 1819 (75.3) |
| Divorced/Separated | 6 (0.6) | 20 (1.5) | 26 (1.1) |
| Widowed | 21 (1.9) | 231 (17.4) | 252 (10.4) |
| **Occupation** | | | |
| Government employee | 42 (3.9) | 19 (1.4) | 61 (2.5) |
| Non-Government employee | 40.6 (37.4) | 184 (13.8) | 590 (24.4) |
| Self employed | 403 (37.1) | 183 (13.8) | 586 (24.3) |
| Student | 56 (5.2) | 29 (2.2) | 85 (3.5) |
| Homemaker | 10 (0.9) | 806 (60.6) | 816 (33.8) |
| Retired | 57 (5.2) | 14 (1.1) | 71 (2.9) |
| Unemployed (able to work) | 49 (4.5) | 34 (2.6) | 83 (3.4) |
| Unemployed (Unable to work) | 63 (5.8) | 60 (4.5) | 123 (5.1) |

Prevalence of obesity 46.1% (95% CI: 44–48.1) was more than twice higher in the study population when compared to overweight 17.6% (95% CI: 16.1–19.1). One-third 33.6% (95% CI: 31.6–35.5) of the population had hypertension that was significantly higher among men and urban residents. DM was present in one-fourth 26.7% (95% CI: 24.1–29.1) of the population, with significantly higher prevalence among men. Over one-third 34.8% (32.5–37.5) had hypercholesterolemia that was significantly higher among urban population (Table 3).

Socio-demographic determinants of various risk factors are given in Table 4. The probabilities of tobacco and alcohol use were significantly higher among men by 26% (PR: 1.26, 95% CI: 1.23–1.3) and 49% (PR: 1.49, 95% CI: 1.4–1.54) respectively. Both these risk factors declined with increasing levels of education. Physical inactivity was significantly higher among urban population (PR: 1.24, 95% CI: 1.17–1.31), while showing gradual increase in its prevalence with increasing levels of education. Hypertension (PR: 0.9, 95% CI: 0.86–0.95) and DM (PR: 0.9, 95% CI: 0.84–0.96) were significantly lower among women by about 9%. Overweight and obesity also increased steadily across increasing education levels.

**Table 2. Mean (confidence interval) of various behavioral and physical measurements and biochemical risk factors by age group, gender and residence among the study participants (N = 2415).**

| Variables | Mean (CI) of behavioral risk factors on an average per day | | | |
|---|---|---|---|---|
| | Daily tobacco use (smoke form) | Standard drinks of alcohol consumed | Servings of fruits and vegetables intake | MET of physical activity/week |
| **Age (in years)** | | | | |
| 18–44 | 6.65 (5.47–7.83) | 4.97 (4.5–5.4) | 3.1 (3.0–3.2) | 1697 (1532–1862) |
| 45–69 | 11.64* (9.6–13.7) | 6.33* (5.7–6.9) | 2.8* (2.7–2.9) | 1762 (1600–1925) |
| **Gender** | | | | |
| Male | 9.55* (8.2–10.9) | 5.66 (5.3–6.0) | 3.1 (3.0–3.2) | 2068 (1879–2257) |
| Female | - | 3.25 (0–6.53) | 2.8* (2.7–2.9) | 1454* (1313–1594) |
| **Residence** | | | | |
| Urban | 10.2 (8.5–12.0) | 5.7 (5.3–6.2.0) | 3.2 (3.1–3.3) | 1371* (1239–1504) |
| Rural | 7.9 (6.0–10) | 5.5 (4.8–6.2.0) | 2.5* (2.4–2.7) | 2551 (2333–2769) |
| **Overall** | 9.58 (8.24–10.92) | 5.63 (5.25–6.0) | 2.94 (2.9–3.0) | 1730 (1614–1845) |
| | **BMI** | **Waist Circumference** | **Systolic Blood Pressure** | **Diastolic Blood pressure** |
| **Age (in years)** | | | | |
| 18–44 | 25 (24.7–25.5) | 90.6 (89.9–91.2) | 120 (119–121) | 79.8* (79.0–80.4) |
| 45–69 | 25 (24.9–25.0) | 91 (90.0–91.7) | 128* (127–129) | 82.9 (82.0–83.5) |
| **Gender** | | | | |
| Male | 24.3 (23.9–24.7) | 88.6 (88.0–89.0) | 127* (126–128) | 83.6* (83.0–84.3) |
| Female | 25.8* (25.5–26.0) | 92.6* (91.9–93.3) | 122 (121–123) | 79.5 (79.0–80.0) |
| **Residence** | | | | |
| Urban | 25.5 (25.0–26.0) | 92* (91.5–92.5) | 125* (124–126) | 82* (81.4–82.5) |
| Rural | 24.3 (23.9–34.6) | 87 (86.0–88.0) | 122 (121–123) | 80 (79.0–81.0) |
| **Overall** | 25.1 (24.9–25.4) | 90.8 (90.0–91.3) | 124.3 (123.6–125) | 81.4 (81.0–81.8) |
| | Mean (Confidence Interval) of biochemical risk factors | | | |
| | **Fasting Blood Glucose** | | **Total Cholesterol** | |
| **Age (in years)** | | | | |
| 18–44 | 105 (101–110) | | 174 (171–177) | |
| 45–69 | 119* (115–124) | | 188* (184–191) | |
| **Gender** | | | | |
| Male | 118 (112–123) | | 179 (175–183) | |
| Female | 110 (105–113) | | 183 (180–186) | |
| **Residence** | | | | |
| Urban | 116 (112–120) | | 184* (181–187) | |
| Rural | 106 (100–112) | | 175 (171–180) | |
| *Overall* | *113 (110–116)* | | *182 (179–184)* | |

*Indicates statistically significant difference between the sub-groups

## Discussion

The study revealed a marginal increase (12.3%) in tobacco use in Puducherry district in comparison to the Global Adult Tobacco Survey (GATS-2, 2016–17) finding (11.2%) for the district [20]. The increase in tobacco use was much pronounced among males (17.7% in GATS to 24.4%), and in urban areas (7.5% in GATS to 12.7%). On the other hand, tobacco use among women has shown a considerable decline (5.1% in GATS to 2.3%) [20]. The surge in smoked tobacco in the district raises concerns, as Puducherry has shown a substantial decline in tobacco use during the last decade (GATS 1 vs 2) [20, 21]. Recent program initiatives such as formation of district-level coordination committee and field task force shall provide the

**Table 3. Prevalence of various risk factors of NCDs by age group, gender and residence among the study participants.**

| Variables | Current tobacco use, % (CI) | Current use of alcohol, % (CI) | Harmful use of alcohol, % (CI) |
|---|---|---|---|
| **Age (in years)** | | | |
| 18–44 | 11.5 (9.9–13.4) | 19.1 (16.7–21.3) | 8.8 (7.2–10.5) |
| 45–69 | 13 (11.1–15) | 17.9 (15.7–20) | 10.3 (8.6–12) |
| **Gender** | | | |
| Male | 24.4* (21.7–26.9) | 40.6* (37.5–43.9) | 21* (18.7–23.5) |
| Female | 2.3 (1.6–3.2) | 0.4 (0.1–0.8) | 0.2 (0–0.5) |
| **Residence** | | | |
| Urban | 12.7 (11.1–14.4) | 18.1 (16.4–20.1) | 9.5 (8–10.9) |
| Rural | 11.3 (9.1–13.8) | 19.2 (16.1–22.1) | 9.7 (7.5–11.7) |
| **Overall** | 12.3 (11–13.7) | 18.5 (16.9–20) | 9.6 (8.4–10.8) |
| | **Low physical activity, % (CI)** | **Inadequate intake of fruits and vegetables, % (CI)** | **Hypertension, % (CI)** |
| **Age (in years)** | | | |
| 18–44 | 48.6 (45.6–51.5) | 83.7 (81.7–85.8) | 20.7 (18.7–23.3) |
| 45–69 | 42.9 (40.1–45.6) | 89.8* (88.1–91.5) | 46.5* (43.7–49.5) |
| **Gender** | | | |
| Male | 41.5 (38.5–44.5) | 83.4 (81.1–85.6) | 37.1* (34.3–40.1) |
| Female | 49.2* (46.6–51.8) | 89.5* (87.9–91) | 30.8 (28.4–33.2) |
| **Residence** | | | |
| Urban | 52.9* (50.7–55.5) | 85.5 (83.8–87.2) | 36* (33.8–38.5) |
| Rural | 29.3 (26–32.6) | 89.8* (87.6–91.8) | 28.2 (25.1–31.3) |
| **Overall** | 45.8 (43.8–47.8) | 86.8 (85.4–88.1) | 33.6 (31.6–35.5) |
| | **Overweight, % (CI)** | **Obesity, % (CI)** | **Abdominal obesity, % (CI)** |
| **Age (in years)** | | | |
| 18–44 | 16.8 (14.6–19) | 45.1 (42.4–48) | 67.4 (64.8–70) |
| 45–69 | 18.3 (16.4–20.6) | 47 (44.2–49.9) | 69.1 (66.4–71.7) |
| **Gender** | | | |
| Male | 21.7* (19.4–24.2) | 37.5 (34.7–40.2) | 46 (43.1–49.2) |
| Female | 14.2 (12.4–15.9) | 53.1* (50.4–56) | 86.5* (84.6–88.3) |
| **Residence** | | | |
| Urban | 18.5 (16.6–20.3) | 49.2* (46.8–51.6) | 71.9* (69.7–74) |
| Rural | 15.6 (13–18.3) | 38.9 (35.3–42.4) | 59.8 (56.2–63.6) |
| **Overall** | 17.6 (16.1–19.1) | 46.1 (44–48.1) | 68.2 (66.4–70.1) |
| | | **Diabetes Mellitus, % (CI)** | **Hypercholesterolemia, % (CI)** |
| **Age (in years)** | | | |
| 18–44 | | 16.5 (13.4–19.7) | 24.4 (20.9–27.9) |
| 45–69 | | 35.3* (31.8–38.9) | 43.7* (39.4–47.6) |
| **Gender** | | | |
| Male | | 31.6* (27.5–35.7) | 34.6 (30.5–39.3) |
| Female | | 23.2 (20.1–26.6) | 35 (31.6–38.6) |
| **Residence** | | | |
| Urban | | 27.6 (24.5–30.9) | 37.9* (34.7–41.1) |
| Rural | | 24.4 (20.1–29.1) | 27.3 (22.7–32.0) |
| **Overall** | | 26.7 (24.1–29.1) | 34.8 (32.5–37.5) |

*Indicates statistically significant difference between the sub-groups

**Table 4. Determinants of various risk factors of NCDs in the population of Puducherry district (N = 2415).**

| Variables | Current tobacco use | Current alcohol use | Physical Inactivity | #Insufficient F&V intake | Hypertension | Diabetes mellitus | Overweight /obesity | Raised Cholesterol |
|---|---|---|---|---|---|---|---|---|
| | aPR (95% CI) | aPR (95% CI) | aPR (95% CI) | aPR (95% CI) | aPR (95% CI) | aPR (95% CI) | aPR (95% CI) | aPR (95% CI) |
| **Gender** | | | | | | | | |
| Male | 1.26* (1.23–1.3) | 1.49* (1.4–1.54) | 1 | 1 | 1 | 1 | 1 | 1 |
| Female | 1 | 1 | 1.04 (0.98–1.09) | 1.06* (1.03–1.1) | 0.9* (0.86–0.95) | 0.9* (0.84–0.96) | 1.02 (0.97–1.08) | 1.04 (0.95–1.13) |
| **Residence** | | | | | | | | |
| Rural | 1 | 1 | 1 | 1 | 1 | 1 | 1 | 1 |
| Urban | 1.03 (0.99–1.06) | 1.01 (0.98–1.04) | 1.24* (1.17–1.31) | 0.96 (0.92–1) | 1.09* (1.05–1.14) | 1.03 (0.97–1.09) | 1.12* (1.07–1.17) | 1.1* (1.03–1.2) |
| **Age (in years)** | | | | | | | | |
| 18–44 | 1 | 1 | 1 | 1 | 1 | 1 | 1 | 1 |
| 45–69 | 0.97 (0.94–1) | 0.95* (0.92–0.98) | 0.99 (0.93–1.04) | 1.04 (1–1.08) | 1.22* (1.16–1.27) | 1.13* (1.06–1.2) | 1.03 (0.99–1.08) | 1.19* (1.11–1.27) |
| **Education** | | | | | | | | |
| No formal education | 1 | 1 | 1 | 1 | 1 | 1 | 1 | 1 |
| Primary education | 0.96 (0.92–1) | 0.99 (0.96–1.03) | 1.09* (1.01–1.17) | 1.01 (0.9–1.04) | 1.04 (0.97–1.11) | 1.04 (0.95–1.13) | 1.1* (1.02–1.18) | 1.03 (0.94–1.14) |
| Secondary education | 0.92* (0.88–0.97) | 0.93* (0.88–0.96) | 1.14* (1.05–1.23) | 0.99 (0.95–1.03) | 0.96 (0.9–1.04) | 1.01 (0.9–1.1) | 1.13* (1.05–1.23) | 1 (0.9–1.13) |
| Graduation and above | 0.87* (0.82–0.93) | 0.89* (0.84–0.94) | 1.25* (1.14–1.37) | 0.97 (0.92–1.03) | 0.97 (0.9–1.05) | 0.98 (0.88–1.09) | 1.2* (1.09–1.3) | 0.96 (0.85–1.09) |
| **Marital status** | | | | | | | | |
| Currently married | 1 | 1 | 1 | 1 | 1 | 1 | 1 | 1 |
| Never married | 1.03 (0.98–1.09) | 1.02 (0.95–1.08) | 0.94 (0.88–1.01) | 0.97 (0.92–1) | 0.89* (0.84–0.94) | 0.85* (0.78–0.91) | 0.8* (0.75–0.86) | 0.97 (0.88–1.07) |
| Widowed/divorced | 1.07* (1.02–1.12) | 0.99 (0.96–1.02) | 1 (0.92–1.09) | 1 (0.96–1.04) | 1.06 (0.9–1.14) | 1.06 (0.97–1.16) | 1 (0.93–1.07) | 0.99 (0.9–1.1) |
| **Occupation** | | | | | | | | |
| Home maker | 1 | 1 | 1 | 1 | 1 | 1 | 1 | 1 |
| Government employee | 0.98 (0.92–1.05) | 0.98 (0.89–1.09) | 0.85* (0.73–0.98) | 0.94 (0.84–1.06) | 1.04 (0.91–1.18) | 1.1 (0.94–1.3) | 1.04 (0.93–1.17) | 1.01 (0.8–1.2) |
| Non-government employee | 1.01 (0.98–1.04) | 1.02 (0.98–1.05) | 0.93* (0.86–0.99) | 1.04* (1.01–1.08) | 0.99 (0.93–1.04) | 0.99 (0.92–1.07) | 0.93* (0.87–0.98) | 1.09 (1–1.19) |
| Self-employed | 1.03 (0.99–1) | 1.05* (1.02–1.09) | 0.91* (0.85–0.97) | 1.01 (0.97–1.04) | 0.97 (0.91–1.03) | 0.95 (0.88–1.03) | 0.93* (0.88–0.99) | 1.05 (0.96–1.14) |
| others | 0.96 (0.92–1) | 0.93* (0.89–0.97) | 0.95 (0.9–1) | 1.03 (0.98–1.08) | 1.01 (0.94–1.07) | 1.12* (1.02–1.23) | 0.86* (0.81–0.93) | 1 (0.92–1.1) |

*Significant results: p≤0.05

#Insufficient intake of fruits and vegetables, aPR: Adjusted Prevalence Ratio, CI: Confidence Interval

necessary impetus for better planning, coordination, implementation and monitoring the Cigarettes and Other Tobacco Products Act (COTPA) violations in the district. Evidence from low-and-middle income countries presenting similar prevalence rates as in this study, showed that despite exerting consistent tobacco control efforts, the countries have failed to control the tobacco menace over time, in terms of tobacco induced morbidity and mortality [22]. This observation points to the district tobacco control authorities to further strengthen the tobacco control policies and implementation in order to drive down the rising prevalence signaled from this study as well as its consequences.

Over one-third (40.4%) of men in Puducherry district were current alcohol user. This was comparable to the National Family Health Survey-4 (NFHS) prevalence (41%) for the district [23], but higher than the national prevalence (29.2%) [24]. The observed prevalence was less compared to the neighbouring state of Tamilnadu (46.7%), but higher than Kerala (37%) and Karnataka (29.3%) [25]. Studies carried out using STEPS methodology in other Indian states showed prevalence between 31.1% and 40% [26–28]. NFHS-4 was done among the 15–45 years age group and only a minority were men (14%) which tend to underestimate the alcohol prevalence for the district [25]. Considering this, the alcohol prevalence observed in the study that is comparable to NFHS-4 finding indicates the rising levels of alcohol use in the district. This carries an important public health implication, as it has been well established that South Asians taking more than moderate level of alcohol have increased risk for Coronary Artery Diseases [29]. In comparison to the global trends, the prevalence of current alcohol use observed in the study is higher compared to the neighboring Asian and Middle-east countries, but substantially lower than Australia, Western Europe, and north American countries [30].

Nearly half (45.8%) of study population was physically inactive and it was significantly higher among women (49.2%) and urban areas (52.9%). This prevalence was much lesser when compared to Tamilnadu (65.8%), a neighbouring state having the population socio-demographic characteristics same as this study [6]. Similar to the current finding, a recent study from Puducherry also reported physically inactivity among half (49.7%) of urban population [31]. Another study conducted across seven states in India in the year 2007 found differential levels of physical inactivity ranging between 42.3% and 81.2%. These findings although evolved from varied regions, they clearly implied a markedly higher prevalence among both men (38.8%) and women (46.1%) in India [6]. The wide-ranging prevalence also signifies the need for state-specific determinants to implement evidence-based population-specific interventions to promote physical activity in the country. Considering that physical inactivity is significantly higher among urban population, and increasing urbanization of rural areas in the years ahead [32], future studies needs to examine the extent of pedestrian-friendly neighbourhoods, public access to parks, open spaces, public transport and other social amenities across states, which are proven to improve physical activities at population level [33]. The physical inactivity levels observed from this study (45.8%) is substantially higher when compared to several country-wide STEPS survey reports from South-East Asian and African countries [34–40].

Substantiating the higher levels of physical inactivity, two-third (63.7%) of the study population was overweight or obese. This was twice higher compared to NFHS-4 finding (36%) for the district [23]. The urban prevalence (67.7%) of overweight or obesity was also higher as compared to a more recent survey (64.4%) from Puducherry [41]. The prevalence of overweight/obesity estimated in the study is apparently higher in comparison to those reported from neighbouring South-East Asian countries that ranged between 12.1% (Timor Leste) and 39.2% (Bhutan). The apparently lower prevalence could be because the current study used BMI classification for Asian Indian, whereas the country wide STEPS surveys used the WHO classification [34–37, 42]. The markedly increased prevalence of this risk factor observed in the study calls for immediate population-level obesity prevention and control programs to target women, urban residents, and school children of Puducherry district. This public health action gains importance because evolving evidences predict doubling of overweight, and tripling of obesity levels in the country by 2040 [43] which are mainly linked to sedentary and changing dietary habits across populations.

In India, dietary risk factor is the leading cause for morbidity and mortality among the states experiencing high epidemiological transition like Puducherry [2]. In congruence, nearly nine out of 10 people (86.8%) in Puducherry district consumed less than recommended

amount of fruits and vegetables. A survey from urban Puducherry also reported much higher prevalence (97.5%) as compared to the current study finding (85.5%) [41]. India, despite being the second-largest producer of fruits and vegetables [44], low intake of fruits and vegetables is highly prevalent across Indian states ranging from 76% in Maharashtra to 99% in Tamilnadu [6]. Similar higher prevalence was also reported by NFHS-3, while showing decreasing levels of intake with decreasing economic status [45]. Higher levels of inadequate intake of fruits and vegetables observed in the study is also consistent with findings from other low-and-middle income countries [34, 36, 37, 42, 46]. Addressing this situation requires multi-sectoral, multi-level and multi-pronged interventions to promote indigenous fruits and vegetables, and streamline its supply chain, which are crucial to enhance 'availability and affordability' of fruits and vegetables across Indian states.

One-third of population in Puducherry district had hypertension (33.6%) that was significantly higher among men (37.1%) and urban population (36%). This finding was almost three times higher when compared to NFHS-4 finding for the district (12.25%) [23], and higher than national (29.8%) and pooled prevalence for south India (26.4%) [47]. The prevalence was also higher compared to several African and south-east countries [34, 36, 38–40, 46] expect for Timor Leste (39.3%) and Bhutan (35.7%) which were comparable to the current study finding [35, 42]. The urban prevalence (36%) observed was also higher compared to a recent study (32.5%) from urban Puducherry [41]. The large discordance in the prevalence, in comparison to NFHS-4 shall be attributed to relatively younger population (15–45 years) and over-representation of women (86%) in the later survey [25]. Significantly higher prevalence ratio among men and 45–69 years age group in the current study also substantiates the observed discordance with NFHS-4 finding. Higher prevalence among men and urban population were also reported in other Indian studies [26].

Diabetes is the leading metabolic risk factor in claiming highest disability adjusted life years (DALYs), in those Indian states experiencing high epidemiological transition [48] like Puducherry. Prevalence of diabetes in the study (26.7%) was substantially higher than the national prevalence of India (7.3%) [7] and other south-east Asian countries like Sri Lanka (7.4%), Bhutan (6.4%), Myanmar (5.9%), Bangladesh (4%) and Nepal (3.6%) [34–37, 46]. Its prevalence was also higher compared to socio-economically comparable India states like Tamilnadu (17.5%), Kerala, (14.8%) Punjab (15.5%). The higher prevalence reported in the current study needs to be interpreted in relation to the time points when those compared studies were conducted. Because the prevalence of diabetes in India has increased by 1.78 times over the last decade across Indian states which might have corresponded to higher prevalence in the current study [49]. Further, a large scale study from Tamilnadu showed that during the decade from 2006 to 2016 the prevalence of diabetes had increased marginally in cities by 8%, but significantly in towns (39%) and villages (34%) [50]. This also substantiates the higher prevalence observed, especially in the rural population of Puducherry (24.4%) which was comparable to several urban populations of the country.

Hypercholesterolemia is one of the key modifiable risk factors for cardiovascular diseases. Its prevalence in Puducherry district (34.8%) was significantly higher compared to the prevalence from four Indian states (13.9%) that ranged between 4.3% (Jharkhand) and 18.3% (Tamilnadu) [51]. The current study finding was also higher compared to a recent study from Tamilnadu (31.4%) [52]. The difference observed between the four states and current study could be due to the methodological differences, as the former survey assessed cholesterol levels among every fifth participant [51]. However, the regional disparity presented across states reaffirms the need for state-specific estimates and determinants in the country which are crucial for devising community-based targeted interventions such as screening, dietary advice, improving physical activity across populations.

The prevalence of various NCDs and its risk factors in the study population was alarming, yet not unanticipated, given the established findings from the states experiencing similar epidemiological transition, as Puducherry [26, 28, 50, 52, 53]. Recent evidences also presents a positive correlation between urbanization and prevalence of NCD risk factors such as physical inactivity, raised BMI, and low intake of fruits and vegetables [54].

In cognizant of the rising burden, several key policies and programs have been adopted in the country since the last decade. About thirteen national programs that directly or indirectly contribute towards prevention and control of NCDs have been rolled out during this period, with NPCDCS as the central to the NCD activities in the country [55]. In 2013, India became the first country to adopt the Global monitoring framework for major NCDs and has set a target of 25% relative reduction in premature NCDs deaths by 2025 [4]. The ten targets and 21 indicators outlined in the National NCD monitoring framework provided the necessary direction to devise strategies to tackle the rising NCDs and its risk factors in the country [56]. However, the country is facing major challenges in the progress towards achieving the targets such as inadequate resources (manpower, financial and technical constrains) and lack of political commitment causing ineffective implementation of various program activities across the states [55].

For instance, an assessment of NPCDCS activities in the state of Karnataka revealed that about one third (31%) of Auxillary Nurse Midwife remained untrained under the program who form the back bone for conducting population-based screening for NCDs and its risk factors [57]. Similarly, during the years 2016–17, the states of Punjab and Haryana have spent only about 0.6% to 1% of the respective total state health mission's budget for health promotion activities such as Information Education Communication/Behavior Change Communication. This reflects poor focus on health promotion activities which are crucial in preventing the development of these risk factors in populations [58].

Various studies conducted across states in the country using STEPS or STEPS aligned methodology showed wide ranging as well as considerably high prevalence of various NCDs and its risk factors. (Table 5) Considering the diverse nature of NCD prevalence, conducting state/region specific NCD survey is crucial to devise population specific and individual-tailored interventions. But, except for the states of Kerala, Punjab and Haryana, state specific estimates of NCDs and its risk factors are currently unavailable. The lack of baseline estimates across states also makes it difficult for states to gauge the effectiveness of various interventions implemented under the NPCDCS program. This calls for the required impetus at the policy front for institutionalizing the NCD surveillance activities across states in the country.

Another major challenge in addressing the rising NCDs and its risk factors is that various departments other than health remain insensitive about their role in prevention of NCDs in the population. Therefore, in line with the national multi-sectoral action plan, strengthening multi-sectoral partnership and undertaking joint prevention activities could play a significant role in preventing the development of NCD risk factors in the population [59]. For example, the food processing sector shall be persuaded to reduce salt content in the foods, and the departments of urban planning and transport shall redesign the available infrastructure for transportation so as to improve physical activity in the population.

## Limitations

Interview based nature of the study could have underestimated a few behavioral risk factors due to socially desirable responses. Behavioral risk factors assessed depended on participant's ability to recall their health behaviors which might have led to recall bias. However, interviews conducted by trained investigator and use of standardized STEPS questionnaire and show

**Table 5. Summary of WHO STEPS surveys from India.**

| Author & year | Geographic Region | State | Current tobacco use (%) | Smoked tobacco (%) | Smokeless tobacco (%) | Current alcohol use (%) | Physical Inactivity (%) | Overweight/ Obesity (%) | *HTN (%) | #DM (%) |
|---|---|---|---|---|---|---|---|---|---|---|
| Selvaraj et al. (2014) [41] | South | Puducherry | 8.1 | NR | NR | 12.2 | 15.7 | 64.4 | 32.5 | 10.3 |
| Oommen et al. (2010) [62] | South | Tamilnadu | 20.5 | NR | NR | 24.3 | 50.8 | 39.7 | 21.4 | 15.9 |
| Vijayakarthikeyan et al. (2017) [63] | South | Tamilnadu | 17.8 | 15.4 | 2.4 | 17.3 | 50.2 | 49.5 | 15.7 | 21.9 |
| Saju et al. (2019) [64] | South | Kerala | NR | 13.6 | 4.1 | 22.1 | NR | NR | 26.3 | 19.1^ |
| Harikrishnan et al. (2011) [65] | South | Kerala | 17.8 | NR | NR | 21.2 | NR | 38.7 | 28.9 | 15.6 |
| Thankappan et al. (2010) [26] | South | Kerala | 21.9 | NR | NR | 11.1 | 6.9 | 30.7 | 28.8 | 14.8 |
| Sathish et al. (2010) [66] | South | Kerala | NR | 16.7 | 18.3 | 22.6 | 82.2 | 41.5 | 36.6 | NR |
| Tondare et al. (2010) [67] | South | Karnataka | 29.93 | NR | NR | 27.29 | NR | NR | 13.3 | NR |
| Kokane et al. (2007) [68] | Central | Madhya Pradesh | 34.2 | NR | NR | 4.5 | 19.6 | 18.7 | 22.3 | 6.3 |
| Basu et al. (2013) [69] | East | West Bengal | 22.9 | 12.3 | 40 | 8.1 | 38.5 | NR | NR | NR |
| Pitchai et al. (2017) [70] | West | Maharashtra | NR | 20.4 | 19.9 | 15.8 | 52.2 | NR | 23.1 | 15 |
| Srivastav et al. (2013) [71] | North | Uttar Pradesh | NR | 15.3 | 25.2 | 8.4 | 14.2 | 39.9 | 17.8 | 10.3 |
| Agarwal et al. (2016) [72] | North | Uttar Pradesh | 34.3 | 40.9 | 48.5 | 12.7 | 88 | 15.7 | 29.7 | NR |
| Agarwal et al. (2017) [73] | North | Uttar Pradesh | NR | 26.2 | 27.08 | 24.1 | NR | 34.86 | NR | NR |
| Thakur et al. (2016) [9] | North | Haryana | 23.5 | NR | NR | 10.5 | 16 | 35.2 | 26.2 | NR |
| Thakur et al. (2014) [15] | North | Punjab | 11.3 | 7.2 | 5.2 | 15 | 31 | 41.4 | 40.1 | 14.3 |

*HTN: Hypertension

#DM: Diabetes Mellitus, NR: Not Reported

^Assessment based on self-report

cards to facilitate recall of health risk behaviors helped to minimize these biases. In the study, physical inactivity levels were determined using GPAQ questionnaire—a globally accepted scale. But, emerging evidences raises concerns over underestimation of household and leisure kind of physical activities if GPAQ is used [60, 61]. This possibly explains the higher physical inactivity levels observed among women in the study.

## Conclusion

The study highlights higher prevalence of various behavioral and biological risk factors of NCDs including its determinants for the district of Puducherry. Notably, one third and one-fourth men in the district were tobacco and alcohol users, respectively. About half of the population was physically inactive (45.8%) and nine out of ten people (86.8%) were taking unhealthy diet. These behavioral risks corresponded to higher levels of metabolic risk factors by showing generalized (46.1%) and abdominal obesity (68.2%) in half of the population and hypertension and diabetes mellitus in one-third (33.6%) and one-fourth (26.7%) of the

population respectively. In the population, increasing education levels decreased tobacco and alcohol but raised physical inactivity and obesity. The higher prevalence of these risk factors, beyond serving as the baseline estimates for NCD risk factors in the district, underscores the urgent need for multi-sectoral, population based health promotion interventions which are indispensable for rendering effective prevention and control of NCDs in the district of Puducherry.

## Author Contributions

**Conceptualization:** Jayaprakash Sahoo, Subitha Lakshminarayanan, Zachariah Bobby, Sitanshu Sekhar Kar.

**Data curation:** Parthibane Sivanantham, Sitanshu Sekhar Kar.

**Formal analysis:** Parthibane Sivanantham.

**Funding acquisition:** Sitanshu Sekhar Kar.

**Investigation:** Parthibane Sivanantham.

**Methodology:** Parthibane Sivanantham.

**Project administration:** Parthibane Sivanantham.

**Software:** Parthibane Sivanantham.

**Supervision:** Jayaprakash Sahoo, Subitha Lakshminarayanan, Zachariah Bobby, Sitanshu Sekhar Kar.

**Validation:** Jayaprakash Sahoo, Subitha Lakshminarayanan, Sitanshu Sekhar Kar.

**Visualization:** Parthibane Sivanantham, Jayaprakash Sahoo, Subitha Lakshminarayanan, Zachariah Bobby, Sitanshu Sekhar Kar.

**Writing – original draft:** Parthibane Sivanantham.

**Writing – review & editing:** Parthibane Sivanantham, Jayaprakash Sahoo, Subitha Lakshminarayanan, Zachariah Bobby, Sitanshu Sekhar Kar.

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
