## [Decision Letter · Decision Letter 0]

22 Oct 2020

PONE-D-20-21171

Profile of risk factors for Non-Communicable Diseases (NCDs) in a highly urbanized district of India: Findings from district-wide STEPS Survey

PLOS ONE

Dear Dr. Kar,

Thank you for submitting your manuscript to PLOS ONE. After careful consideration, we feel that it has merit but does not fully meet PLOS ONE’s publication criteria as it currently stands. Therefore, we invite you to submit a revised version of the manuscript that addresses the points raised during the review process.

We look forward to receiving your revised manuscript.

Kind regards,

Denis Bourgeois

Academic Editor

PLOS ONE

Journal Requirements:

3.  Please provide details regarding the pilot testing of the questionnaire within the methods section

Reviewers' comments:

Reviewer's Responses to Questions

**Comments to the Author**

1. Is the manuscript technically sound, and do the data support the conclusions?

Reviewer #1: Yes

Reviewer #2: Yes

2. Has the statistical analysis been performed appropriately and rigorously? 

Reviewer #1: Yes

Reviewer #2: Yes

3. Have the authors made all data underlying the findings in their manuscript fully available?

Reviewer #1: Yes

Reviewer #2: Yes

4. Is the manuscript presented in an intelligible fashion and written in standard English?

Reviewer #1: Yes

Reviewer #2: Yes

5. Review Comments to the Author

Reviewer #1: This study was conducted because no systematic survey of risk factors for Non-Communicable Diseases has been conducted in Puducherry State, which is one of the most urbanized states in India.

Study setting and design, sample size calculation, sampling method, data collection instrument and data collection procedure (for steps 1, 2, 3) , statistical analysis are very well explained and appear to follow exactly the STEPwise Approach to Surveillance (STEP) methodology prescribed by the World Health Organization (WHO).

Ethics approval has been obtained. The WHO STEPS questionnaire (version 3.2) was culturally adapted, translated into Tamil and tested before data collection. The results are well presented and the tables are very clear and understandable. Subgroup differences are well described and highlighted. Key results are discussed and compared with a relevant bibliography. Limitations and strengths are commented. Finally some policies and some urgent actions that should derive from the findings are suggested.

It is a very relevant study, well conducted and transcribed in accordance with the recommendations of WHO.

I think this study will be very useful for the preventive policy to be implemented in the district of Puducherry.

I only have a few questions and suggestions for the authors:

- Line 81: what is 9.8 lakhs?

- You should indicate the exclusion criteria

- Did the WHO STEPS team receive a copy of the STEP instrument after adaptation and before finalization as recommended?

- Regarding the recruitment and training of staff: specify the number of field team members and briefly explain the training of data collectors

- Line 141: specify the room and location chosen for the collection of blood samples in the STEP 3 procedure

Reviewer #2: Please specify the year of study in the title.

Please specify the period of study in the text.

Specify the name of the institute to which the ethics committee belongs.

Unless I'm mistaken, I don't see the survey rate

Refer in the methodology to a study or studies that describe the STEPS protocol in detail.

Bibliographic references are too old (in yellow in the attachment)

Your discussion appears too vague. the discussion is not synthetic enough. Couldn't you synthesize the different step studies from the different regions of India in a table or a figure?

Items can be deleted (in blue in the attachment). It may be necessary to broaden to a public health dimension i.e. recommendations or possible actions in reference to WHO diabetes policies, India, etc...

Briefly broaden the discussion to include other STEP studies published internationally.

6. PLOS authors have the option to publish the peer review history of their article (what does this mean?). If published, this will include your full peer review and any attached files.

Reviewer #1: No

Reviewer #2: No

---

## [Author Response · Author response to Decision Letter 0]

3 Dec 2020

The comments received from the reviewers have been addressed in the manuscript. As per revised submission guidelines, both 'track change file' and 'Manuscript' files have been uploaded.

---

## [Editor Report · Decision Letter 1]

26 Dec 2020

Profile of risk factors for Non-Communicable Diseases (NCDs) in a highly urbanized district of India: Findings from Puducherry district-wide STEPS Survey, 2019-20

PONE-D-20-21171R1

Dear Dr. Kar,

We’re pleased to inform you that your manuscript has been judged scientifically suitable for publication and will be formally accepted for publication once it meets all outstanding technical requirements.

Kind regards,

Denis Bourgeois

Academic Editor

PLOS ONE
---

## [Editor Report · Acceptance letter]

2 Jan 2021

PONE-D-20-21171R1 

Profile of risk factors for Non-Communicable Diseases (NCDs) in a highly urbanized district of India: Findings from Puducherry district-wide STEPS Survey, 2019-20 

Dear Dr. Kar:

I'm pleased to inform you that your manuscript has been deemed suitable for publication in PLOS ONE. Congratulations! Your manuscript is now with our production department. 

Kind regards, 

on behalf of

Professor Denis Bourgeois 

Academic Editor

PLOS ONE